# Emerging Contributions of Pluripotent Stem Cells to Reproductive Technologies in Veterinary Medicine

**DOI:** 10.3390/jdb12020014

**Published:** 2024-05-07

**Authors:** Raiane Cristina Fratini de Castro, Tiago William Buranello, Kaiana Recchia, Aline Fernanda de Souza, Naira Caroline Godoy Pieri, Fabiana Fernandes Bressan

**Affiliations:** 1Department of Surgery, Faculty of Veterinary Medicine and Animal Sciences, University of Sao Paulo, São Paulo 01001-010, SP, Brazil; raianecfratini@usp.br (R.C.F.d.C.); tiagoburanello@usp.br (T.W.B.); kaiana.recchia@usp.br (K.R.); 2Department of Veterinary Medicine, School of Animal Sciences and Food Engineering, University of Sao Paulo, Pirassununga 13635-900, SP, Brazil; alinsouza25@gmail.com

**Keywords:** livestock, PGCs, iPSCs

## Abstract

The generation of mature gametes and competent embryos in vitro from pluripotent stem cells has been successfully achieved in a few species, mainly in mice, with recent advances in humans and scarce preliminary reports in other domestic species. These biotechnologies are very attractive as they facilitate the understanding of developmental mechanisms and stages that are generally inaccessible during early embryogenesis, thus enabling advanced reproductive technologies and contributing to the generation of animals of high genetic merit in a short period. Studies on the production of in vitro embryos in pigs and cattle are currently used as study models for humans since they present more similar characteristics when compared to rodents in both the initial embryo development and adult life. This review discusses the most relevant biotechnologies used in veterinary medicine, focusing on the generation of germ-cell-like cells in vitro through the acquisition of totipotent status and the production of embryos in vitro from pluripotent stem cells, thus highlighting the main uses of pluripotent stem cells in livestock species and reproductive medicine.

## 1. Introduction

The development of complex multicellular organisms, such as mammalians, begins with the fertilization of an oocyte by a sperm cell. For the adequate formation of competent mature and functional gametes from primordial germ cells (PGCs), interactions between the germ cells themselves and somatic cells within the gonads (ovaries and testicles) are required, and, in the case of the sperm, interaction with the epithelial cells that line the lumen of the female genital tract is also required. After fertilization, the totipotent zygote is formed and undergoes sequential cell divisions, and the morula is developed. Shortly afterward, when the blastocyst stage is reached, two distinct areas are morphologically differentiated: the inner cell mass (ICM), which comprises the pluripotent cells of the epiblast (EPI) and hypoblast (HP), and the trophectoderm (TE), which forms the outer layer of the first differentiated cell lineage and later gives rise to the embryonic placenta [1,2,3]. 

However, although the pre-implantation and development mechanisms of the new individual follow similar processes in mammalians, they have mostly been deeply studied and described in mice. In other species, such as pigs, bovines, and equines, genetic and epigenetic changes during this process (from fertilization until the birth of the new individual) are still mostly unrevealed. Today, studies have reported that the embryonic development of these animals shows several similarities with one another and humans [2]. Therefore, to understand each process during this period, current studies are prospecting new technologies to generate embryos or re-create gametogenesis in vitro (in vitro gametogenesis, or IVG). 

The use of assisted reproductive technologies (ARTs) makes it possible to optimize the generation of embryos, which do not necessarily need to be conceived in a uterine environment. In most domestic species, for example, embryo generation has been reported after in vitro embryo production (IVP) protocols, performing in vitro oocyte maturation (IVM), then in vitro fertilization (IVF), and finally, in vitro culture (IVC). However, using ARTs often fails to overcome some barriers, such as epigenetic reprogramming, leading to undesirable results in livestock species [4]. 

The knowledge of embryogenesis in mammals is essential for improving reproductive technologies and preventing early embryonic loss [5]. However, information on pre-implantation embryonic development in livestock (as well as other domestic and wild species) remains restricted due to the limited access to and manipulation of embryos due to their intrauterine development. Recently, the rapid advancement in stem cell research has allowed research groups to use pluripotent stem cells (PSCs) to provide an in vitro study model capable of generating complex structures similar to mammalian embryos (or part of them), called embryoids [6]. In order to study a specific phase of embryonic development, blastoids are created, which are structures similar to blastocysts but derived from stem cells and include extra-embryonic lineages capable of modeling pre-implantation development [7]. The first blastoids were developed by assembling mouse embryonic stem cells (ESCs) with trophoblastic stem cells (TSCs) [8]. Other research groups have also developed in vitro structures similar to in vivo blastocysts through the differentiation and self-organization of extended pluripotent stem cells (EPSCs) [9] or by the fusion of EPSCs with TSCs [10,11].

In addition to embryoids, advancements have led to research into the in vitro creation of gametes, also known as IVG. The creation of properly functioning in vitro gametes not only provides further exploration of the mechanisms of germ cell development but also promotes numerous possibilities in reproductive medicine, particularly for the study of diseases and for the potential generation of healthy offspring from infertile individuals [12].

This review discusses the most relevant biotechnological methods used in veterinary medicine, such as the acquisition of totipotency and the production of embryos in vitro. In addition, we present the use of pluripotency stem cells in the induction of germ cells and germ-cell-like cells in vitro. 

## 2. “Old” and New Technologies for Embryo Production

For decades, in vitro embryos have been generated through IVP protocols in species such as bovines and humans. The objective is to produce embryos in the laboratory through the IVF of oocytes, which may have matured in vitro (well established for cattle, with recent advances for pigs) or in vivo (mainly for mice and humans). 

In livestock species, oocytes can come from slaughterhouse ovaries or can be derived from ex vivo ovum pick up (OPU) [13]. The successful generation of embryos produced in vitro by IVF was described with the birth of the first mammal (rabbit) in 1959 [14]. In the following years, several other reports of IVF followed by healthy offspring were recorded. Whittingham achieved success working with mice in 1968 [15], followed by the birth of the first human baby ten years later [16]. The first reports of success in livestock species were described in porcine [17] and then in bovine species with the birth of an IVF calf in 1981 [18] (Table 1 and Table 2).

Intracytoplasmic sperm injection (ICSI) and somatic cell nuclear transfer (SCNT) are other technologies that allow for the generation of embryos in vitro in large domestic animals [13]. The ICSI consists of fertilization by the direct injection of a single sperm into the ooplasm. In bovines, ICSI is mostly used for research purposes, since IVP protocols are more efficient commercially [20]. The same applies to ICSI in porcine species, where the low success rates are not commensurate with the expense and effort involved, making it unviable for porcine production [21]. Through SCNT, it is possible to generate a reconstructed embryo, where the nucleus of a somatic donor cell is introduced into an oocyte whose own nuclear DNA has been removed. In bovines, the blastocyst rate is between 10 and 40% in SCNT experiments, of which 10 to 30% develop into calves after being transferred to recipients [22]. In porcine species, blastocyst rates from SCNT embryos are 20 to 40% [22,23], but only 1 to 5% of cloned piglets are born [24,25]. 

Unlike in “IVF permissible species”, such as cattle, IVF in horses is difficult to establish and reproduce; currently, assisted reproduction techniques, such as ICSI, have been used as alternatives. Although numerous attempts to establish this technique have been carried out over the years, when fertilization is achieved, the efficiency is very low compared to other species [26]. Palmer et al. (1991) [19] reported the births of foals after in vitro fertilization using in vivo matured oocytes recovered from the dominant stimulated follicle and spermatozoa treated with some molecules. Furthermore, two other studies achieved, for the first time, a high relative rate (+30%) of IVF when they used a modified IVF technique developed in cattle [19,27,28] (Table 2).

Recently, Felix et al. [28] demonstrated that after treating sperm with some substances such as penicillamine, hypotaurine, and epinephrine (PHE) for a few hours, the co-incubation of cumulus–oocyte complexes (COCs) for 6 h yielded 43% fertilization. Presumptive embryo culture yielded 21% blastocysts, and when co-incubated for 3 h with COC, it yielded 90% fertilization with a blastocyst rate of 74%. They achieved fertilization, blastocyst development, and foal production from oocytes recovered via TVA and post-mortem.

In cattle, IVP protocols have been constantly improved. Still, the blastocyst rates routinely reach around 30–40% [29], and when compared to their in vivo counterparts, differences are found in terms of the microvilli [30], lipid content [31], cryoresistance [30], and genetic expression [32]. In porcine species, IVP performance is lower than in bovines, resulting in variable success rates [33]. The blastocysts generated by IVP are less capable of producing pregnancy and have a lower cell number than their ex vivo counterparts [34]. There is a major unresolved problem in porcine IVP, which is the high proportion of polyspermy. Polyspermic embryos are aneuploid, show abnormal cleavage patterns, have reduced ICM growth, and fail to develop to completion, thus impairing the efficiency of IVP [35]. Given the limiting factors of IVP, it is necessary to look for new perspectives that contribute positively to the generation of embryos in vitro and to the understanding of pre-implantation embryonic development to optimize the use of livestock species as models for human diseases and to maximize the use of embryonic technologies for reproduction (Table 2).

**Table 2 jdb-12-00014-t002:** Old and new technologies for mammalian generation of germ cells and embryos in vitro.

Species	Old Technologies	New Technologies	References
Bovine	IVF, OPU, ICSI, and SCNT	Blastoid (EPSCs + TSCs)	[11,18]
Porcine	IVF, ICSI, and SCNT	OLCs, SSCLCs, and PGCLCS	[17,21,22,36,37,38,39]
Equine	IVF and ICSI		[19,26,27]
Mice	IVF	Blastoid (ESCs + TSCs, EPSCs + TSCs, and EPSCs), PGCLCs, xrOVARY, and rTESTES	[8,12,15,40,41,42]
Human	IVF	Blastoid (naïve PSCs, EPSCs, and iPSCs), PGCLCs, and xrOVARY	[7,16,43,44,45,46]

Legend: in vitro fertilization (IVF); ovum pick up (OPU); intracytoplasmic sperm injection (ICSI); somatic cell nuclear transfer (SCNT); extended pluripotent stem cells (EPSCs); trophoblastic stem cells (TSCs); oocyte-like cells (OLCs); spermatogonial stem cell-like cells (SSCLs); embryonic stem cells (ESCs); primordial germ cell-like cells (PGCLCs); pluripotent stem cells (PSCs); induced pluripotent stem cells (iPSCs); xenogeneic reconstituted ovaries (xrOVARY); in vitro reconstitution of the whole male germ cell (rTESTES).

More recently, studies on early embryonic development have described the first embryoids capable of modeling a blastocyst-like structure (or blastoid) formation in mice [8]. Since then, new studies have appeared in other species (human and bovine) [7,11,43,45]. To date, studies regarding domestic and wild animal species have reported live-born offspring. However, murine blastoids and bovine blastoids have been transferred into the uteruses of recipient females, but due to gaps that have not yet been clarified, the blastoids cannot support complete and genuine embryonic development yet [8,11]. For ethical reasons, human blastoids can be cultured for the minimum time necessary to achieve the scientific objective according to the ISSCR guidelines of 2021, but transfer to the uterus of a human or animal host is prohibited [47]. For this reason, researchers have used other species models to understand the gaps in the early embryonic development of humans since these livestock species, such as bovine and porcine species, present more similar characteristics when compared to rodents, especially during the initial development of in vitro embryos [48,49,50] (Table 2).

Regarding the latter, studies using ESCs with TSCs showed that it was possible to form blastoids that were morphologically and transcriptionally similar to embryonic blastocysts. These blastoids did not support complete and genuine embryonic development, but they were able to infer that embryonic inductions are crucial for the development and implantation of the trophectoderm, which recaptures key aspects of uterine implantation [8].

The formation of blastoids has also been reported from the culture of EPSCs. EPSCs are totipotent cells derived from induced pluripotent stem cells (iPSCs) or blastocyst-stage embryos. These cells can be stably cultured for the long term while maintaining the capacity to contribute to the generation of embryonic (Em) and extra-embryonic (ExEm) lineages in vitro and in vivo [51]. In mice, the generation of blastoids from EPSCs occurs through the segregation of cell lineages and self-organization, forming the three first embryonic cell lineages (epiblast, hypoblast, and trophectoderm). They were able to recapitulate the main morphogenetic events during pre-implantation and early post-implantation developments in vitro and have a similar morphology and cell lineage allocation to the mouse blastocyst [9]. 

Another blastoid model was established in mice by culturing EPSCs and TSCs; the cells were able to self-organize into both embryonic and extra-embryonic lineages, resembling late-stage blastocysts [10]. Recent reports show successfully generated human blastoids resembling each other in terms of their morphology, size, number of cells, cellular composition, and allocation of different cell lineages [45,52]. They can be generated to reproduce a specific stage without notable marks of the previous stages through modifications under culture conditions or component cells, bringing new potential to understanding mammalian development [53]. The formation of blastoids in bovines, humans, and mice is shown in Figure 1.

Based on this recent and promising technology, blastoid models in livestock species have been described in bovines. In this species, the blastoids were generated through the simultaneous culture of self-organized EPSCs and TSCs and maintained for more than two weeks in a rotating 3D culture. It was possible to verify the presence of markers characteristic of epiblast (SOX2), hypoblast (SOX17), and trophectoderm (CDX2) lineages, confirming the presence of all blastocyst lineages. In addition, the bovine blastoids were similar in their morphology, size, number of cells, composition, and cell lineage allocation. They were able to produce maternal recognition signals after transfers to recipient cows when the interferon-tau hormone (INF τ) was measured in the animals’ blood [11]. The bovine model opens the door to testing in other species, representing a valuable model for studying early embryonic development and the causes of early embryonic loss.

In other domestic species, such as swine, blastoids have not yet been described in the literature. In vitro embryo production technology in this species consists of IVP protocols and SCNT [54]. Due to the physiological similarities between the porcine species and humans, the use of these reproductive technologies contributes to increasing production efficiency within the pig industry and in the medical field, allowing them to be used as research bio-models [55].

Despite these promising studies, IVP protocols in porcine species still present difficulties in achieving high fertilization rates and the subsequent development of blastocysts in vitro. High rates of polyspermy, problems with oocyte cytoplasmic maturation in vitro, and low rates of embryonic development are the main obstacles encountered [56]. As a consequence of such limitations, there is a growing desire to develop new technologies for this species due to its economic importance, as well as for biomedical purposes. The generation of porcine blastoids using PSCs could symbolize a new milestone for porcine reproduction and represents an important biomedical model for the study of early embryonic development in humans, avoiding ethical issues.

## 3. Production of Competent Gametes in Livestock

The new technologies for producing mature gametes aim, firstly, to primordially contribute to the understanding of the mechanisms and processes during gametogenesis in vivo, including the morphological, genetic, and epigenetic changes in the germline cells during specification, migration, sexual differentiation, start of meiosis, and differentiation into mature gametes. In livestock, these mechanisms are yet to be unraveled. Hence, herein, we highlight the studies that describe the PGCs in vivo and the technologies used to generate these cells in vitro from PSCs or other types of cells in livestock animals.

### 3.1. Germline Cells In Vivo

After fertilization and during the early phases of embryogenesis, germ cells achieve totipotency so that the genetic information is correctly passed on to the next generation, thereby ensuring genetic diversity. In this period, totipotent cells undergo epigenetic reprogramming and convert to the germinal lineage. The PGCs are, therefore, considered the only source for totipotency, reproduction, inheritance, and evolution in different species of animals [57,58,59,60,61,62,63]. 

The PGCs are better understood and characterized in mice [64,65,66], whereas in other domestic and wild species, few studies have described the signaling pathways, where the cells arise, or even the epigenetic and genetic process that occurs until the cells arrive in the ridge [66,67]. In bovines, it has been described that bovine PGCs (bPGCs) can be identified by alkaline phosphatase staining in E18–E39 embryos [68,69]. According to this study, these cells can be observed in the proximal yolk sac, in the hindgut and midgut, and in the genital ridges. A recent study analyzed cattle fetal ovary between days 40 and 90 of gestational development, and it was shown that at 50 days of development, the PGCs were still in the early stage of differentiation, presenting some similarities in the transcriptome between PGCs in both species [70] (Figure 2).

In porcine species, studies have identified that the first porcine PGCs (pPGCs) emerge in embryos at 12–14 days. The cells in this period express OCT4, BLIMP1, and SOX17 [71,72]. In 2017, Kobayashi et al. [67] and colleagues reported a study comparing the porcine and human PGCs; they described the first pPGCs at 11–12 days after fertilization, co-expressing the SOX17, BLIMP1, NANOG, TFAP2C, and OCT4 markers. The pPGCs start their migration toward the genital ridge at 15 days, stage-equivalent to E8.5 in the mouse, and 2–4 weeks in humans. At E18-22, they start to colonize [73], and at E28-42, they proliferate the genital ridges [73,74]. Zhu et al. [63] investigated the conservation of germline development in detail and compared the expression profiles of pPGCs, hPGCs, and cyPGCs (cynomolgus monkey); they showed similar expression profiles of the key germline genes (*SOX17*, *PRDM1 [BLIMP1]*, *TFAP2C*, *NANOS3*, and *DND1*) and pluripotency genes (*NANOG*, *POU5F1*, and *LIN28A*).

More recently, in 2023 [75], Soares et al. analyzed the expression of epigenetic, pluripotent, and germline markers associated with the development and differentiation of pPGCs at different gestational periods (from 24 to 35 days post-fertilization (dpf) and in adults) and with gender. They showed that the formation of the genital ridge from mesonephros at 24 days of gestation continues to develop until 29 days, when differentiation into primitive gonads begins, ending sexual dimorphism at 35 days of gestation. Also, they revealed that, regardless of sex, there is a difference in the expression of the genes OCT4, SOX2, NANOG, STELLA, VASA, PRDM14, and DAZL when different gestational periods or adult tissues are analyzed. In this species, the PGCs exhibit different epigenetic changes during migration until reaching the genital ridge [28,29]. 

Zhu et al. (2021) [63] described that during the pre- and early migratory stages (E11–35 days after artificial insemination), the pPGCs initiated large-scale epigenomic reprogramming, including DNA demethylation, and the cells showed that H3K27me3 was elevated in the migratory stage (E17) and early gonadal (E25) PGCs, but it decreased sharply in the mid- and late gonadal PGCs. In addition, they showed that the pPGCs pre-migration (E14) had a high 5hmC level, and the DNA methylation reached the lowest levels in gonadal pPGCs. However, Soares et al. (2023) analyzed pPGCs from males and females at 24–35 days after fecundation and reported pPGCs that were negative to H3k27me3 and H3k9me3 in the periods of 25/26, 29, and 35 dpf in female embryos, although in adult gonads, some cells are positive for these epigenetic markers. 

Little research describes the development, sexual differentiation, and signal pathways of PGCs in horses [76,77,78]. Curran et al. [79] evaluated the beginning of the emergence of PGCs in horses. They demonstrated that at 20 days of gestation, there is a low number of phosphatase-positive ePGCs in the gonads, but that in the period from 22 to 26–28 days, the ePGCs reach the gonadal crest and proliferate, demonstrating an increase from 4% to 28% in the number of FA-positive cells in the gonadal ridge. Another study showed that the gonadal crest is undifferentiated at 40 days of gestation [77]. In 2021, Scarlett et al. [78] described that the migration of primordial germ cells from the mesonephros to the gonad was detected at 45 days, suggesting that gonadal sexual differentiation in the horse occurs asynchronously between the sexes and before 45 days of gestation. Furthermore, the number and distribution of ePGCs within the gonad differed between males and females [76,78] (Figure 2).

As already stated, however well described in mice, there might be differences between species in the timescale for each cell–fate transition, developmental synchronicity, and the signaling, transcriptional, and metabolic properties of developmentally homologous cell types [60,80]. For this reason, much has been investigated regarding the developmental mechanisms of such cells in the last decade, and many efforts have been made to reconstitute all these processes in vitro (IVG). One of the technologies used for a complete IVG is the differentiation of pluripotent stem cells (PSCs—induced pluripotent stem cells (iPSCs) or embryonic stem cells (ESCs)) [41,80,81] into mature gametes. This methodology has been used to elucidate the signaling, transcription, and epigenetic regulation processes associated with the specification and development of primordial germ cells (PGCs) [82,83,84] and the mechanisms that cause anomalies or infertility.

### 3.2. X Chromosome Inactivation (XCI)

Sex chromosomes (X and Y) have been present in mammals for approximately 160 million years since before the evolutionary separation of marsupials and placental mammals [85]. The X chromosome contains more than 1000 genes, while the Y chromosome has approximately ten times less than this value, causing an imbalance in the dosage of X-linked genes between females (XX) and males (XY) [86,87]. Thus, to compensate for differences between the sexes, evolution in placental mammals developed the process of inactivating one of the two X chromosome alleles in females [88].

The molecular mechanism for X chromosome inactivation (XCI) begins early in mammalian embryogenesis. One key stage in which XCI is established is during the blastocyst formation. In the pre-implantation stage, during early cleavage divisions, the embryo is a morula, and both X chromosomes in the female embryo are active. As the embryo develops, it undergoes compaction, leading to blastocyst formation. During the blastocyst stage, XCI is initiated, and one of the two X chromosomes in female embryos becomes inactivated. The choice of which X chromosome to inactivate is generally random; some cells may express genes from the maternal X chromosome, while others express genes from the paternal X chromosome. This mosaic pattern of X inactivation can lead to phenotypic diversity among cells and tissues. Once established during the blastocyst stage, the XCI status is maintained as cells divide and differentiate into various cell lineages. The inactivation status of the X chromosome is passed on to daughter cells during cell division. However, not all genes on the X chromosome are subject to inactivation. Some genes escape XCI and remain active on both the active and inactive X chromosomes. In germ cells, X chromosomes are globally reactivated to ensure the proper expression of X-linked genes in the next generation [89]. Also, a form of imprinted XCI occurs in embryonic tissues, such as the placenta. In this process, the paternal X chromosome is preferentially inactivated. 

The initiation of XCI occurs in a region known as the X inactivation center, which produces the specific X inactivation transcript (*XIST*, X-inactive specific transcript). *XIST* is typically expressed from the future inactive X chromosome and plays a crucial role in coating the X chromosome, recruiting chromatin-modifying complexes, and leading to its inactivation [90]. In the context of bovine species, XCI has been less extensively studied than model organisms like mice and humans. XCI in mice commences at the 4-cell stage embryo, specifically involving the exclusive inactivation of the paternal X chromosome (Xp) [91]. However, evidence suggests that the expression of *XIST* RNA in bovine embryos was significantly upregulated at the morula stage [92]. In addition, it is already known that aberrant patterns of XCI can occur during the somatic cell cloning of bovine embryos due to abnormal epigenetic markers [93]. 

X chromosome inactivation is an example of epigenetic regulation that occurs during embryonic development and can also be studied in in vitro culture, for instance, in ESCs and iPSCs [94,95,96]. The process of reprogramming somatic cells into iPSCs involves the resetting of epigenetic marks, including those associated with XCI. During the reprogramming process, the expression of *XIST*, a long non-coding RNA associated with XCI, is generally downregulated. Usually, iPSCs derived from female somatic cells reset the XCI status, leading to the reactivation of one X chromosome. This reactivation is crucial to achieving a pluripotent state and further cell differentiation [97]. The degree of XCI can vary among individual iPSC lines. Even within a single iPSC line, there may be variability in the XCI status among different cells, and it may exhibit a mosaic pattern of X chromosome inactivation. Researchers are aiming to develop iPSCs models that accurately reflect the XCI patterns; this understanding is particularly relevant when using these cells for disease modeling, especially for X-linked disorders.

### 3.3. Gametogenesis In Vitro

In 1980, for the first time, allogenic chimeras were reported using mouse pluripotent stem cells (PSCs) with germline contribution through the injection of these cells into host blastocysts; such a technical achievement was used for decades for several applications in mice. 

More recently, research reports on in vitro gametogenesis performed on mice have been published, and since then, the generation of gametes using PSCs has been reported in humans and mice with different models in vitro [41,66,80,81]. A significant advancement in the field was reported by Hayashi et al. [12,40], showing that PSCs can be induced into epiblast cell-like cells (EpiCLs) and then into PGC-like cells (PGCL-like cells or PGCLCs) with the capacity for complete spermatogenesis and oogenesis and the production of healthy offspring [12,41,98]. Such a successful derivation of PGCLCs in vitro in mice [12,99] opened the possibility of producing functional oocytes or spermatozoa from other species (humans, primates, bovine, and porcine) [38,39,46,61,99]. Recently, the generation of PGCLCs from the iPSCs of the white rhinoceros was reported, with the use of such technology aiming to contribute to the preservation of endangered species [100]. Hopefully, this technique can be used in other species (Figure 3).

In livestock species, such as porcine species, the first studies that showed the generation of in vitro oocyte-like cells (OLCs) were performed by differentiating somatic cells from the skin, adipose, and ovarian tissue [37,38,39]. However, these cells were not analyzed for feasibility or functionality (e.g., the production of live and healthy offspring). In 2021, Pieri et al. described the generation of PGCLCs from iPSCs using a similar model to that described by Hayashi et al. [12] in mice, and the former authors discussed that the resultant porcine PGCLCs presented a similar phenotype to the porcine PGCs in vivo (Figure 3).

The generation of iPSCs from embryonic fibroblasts or non-invasive sources (e.g., urine) from livestock species has been reported [101,102,103,104,105,106], and some of these studies described differentiation into germ-like cells or neuronal-like cells. In 2015, bovine-induced pluripotent stem cells (biPSCs) were induced into bovine PGCLCs (bPGCLCs) through body formation embryoids in the presence of RA and BMP4; however, the production of bPGCLCs was not shown to be efficient since only DDX4 was detected [107]. COSTA et at. [108] reported that bovine skin-derived fibroblasts treated with 2.0 μM 5-azacytidine (5-Aza) for 72 h induced the expression of pluripotency factors (SOX2, OCT4, NANOG, and REX). When the cells were cultured in a differentiation medium supplemented with BMP2, BMP4, or follicular fluid, they showed morphological changes, forming oocyte-like cells and expressing markers for germ cells, meiosis, and oocytes.

In another study, PGCLCs and OLCs were differentiated from ovarian stem cells. According to the authors, after the differentiation period, and after supplementation with BMPs and ovarian fluid in the culture, the bovine cells showed the expression of VASA, DAZL, GDF9, ZPA, and SCP3 [109]. Other studies have also described the generation of sperm-like cells from the co-culture of bovine newborn germ cells with Sertoli cells and retinoic acid [110,111]; however, the generation of sperm-like cells or spermatogonia-like cells (SSCLCs) from pluripotent stem cells has not been reported yet in this species.

In porcine species, the possibility of differentiating porcine somatic cells into iPSCs and then into pPGCLCs has been shown, as has the generation of spermatogonial stem cell-like cells (SSCLs) [38], although the production of mature gametes has not been reported. In a study reported by our group, these cells showed heterogeneity and distinct phenotypic profiles influenced by the in vitro environment. Furthermore, the PGCLCs generated were similar to human and porcine PGCs in vivo [39]. The generation of in vitro oocyte-like cells (OLCs) has been previously reported in pigs after the differentiation of somatic cells from the skin, adipose, and ovarian tissue [36,37,112]; however, these were not analyzed for feasibility or functionality. 

The development of new technologies and protocols, such as the production of PGCLCs and mature gametes through iPSCs, is considered a major technological innovation in the areas of reproduction and biotechnology, as it could promote gains in the production of embryos in vitro, with the production of a greater number of competent oocytes, facilitating the exchange of genetic material. Furthermore, from a feasible perspective, the gene editing of these cells would allow for the genetic improvement of animals or the creation of transgenic animals with better productivity, disease resistance, and commercial value.

## 4. Animal Models and Translational Medicine 

Animal iPSCs play a crucial role in translational medicine, aiming at both human and veterinary applications, as well as in the preservation of genetic material from endangered species, the acquisition of genetically superior animals, and the development of animal products in vitro [113,114]. Large animal iPSCs, such as bovine or porcine, have been used as a biomedical model for human regenerative medicine due to the plasticity that iPSCs have, meaning they have the potential to differentiate into various cell types, mirroring embryonic stem cells [102,115,116]. This versatility makes iPSCs valuable tools for studying gametogenesis and embryo development, reproductive disease modeling, and regenerative medicine applications (Figure 4).

Bovine and porcine species are considered excellent models for biomedical research as they more closely resemble human physiology and anatomy compared to laboratory animals [117,118,119,120,121,122,123]. Thus, biPSCs (bovine iPSCs) or piPSCs (porcine iPSCs) are also being considered for use as a biomedical model for reproductive medicine strategies, embryogenesis, and developmental studies. The porcine model has been used to study the long-term effects of gene-based therapies, the PGCLC process, and the entire gametogenesis. Studies show that the porcine species shares similarities with humans in many of these events [67]. Bovines offers an excellent model to study human in vitro embryo development as they share multiple ovarian diseases, similar gestation periods, and a similar process for germline development [70,119,122,124,125]. Therefore, both species are important when considering the scalability and translatability of regenerative therapies to human patients, like reproductive disorders, to clarify the underlying causes of reproductive challenges in humans and facilitate the development of targeted interventions [122,124]. 

Primordial germ cells (PGCs) derived in vitro from iPSCs have been described in different animal species, including in humans, from infertile patients; however, no further investigations to achieve offspring have been reported in humans [12,39,40,100,105,126,127,128]. Regarding the gametogenesis similarities between bovines and humans, the development of in vitro gametogenesis from bovine iPSCs could enhance our knowledge about the gametogenesis process [70]. Furthermore, PGCs generated from large animal iPSCs would have fewer ethical barriers to generating offspring. Therefore, the development of protocols for in vitro gametogenesis in large mammals, such as bovine and porcine species, from PSCs has not been described in the literature. Our research group has generated bovine and porcine PGCLCs (PGC-like cells) in vitro from iPSCs [39,129] (unpublished data).

In vitro gametogenesis derived from PSCs would improve the understanding of these processes in different species, with the aim to develop viable gametes in a scalable and safe environment with high control to achieve offspring [130], without running into ethical barriers or other challenges, such as the number of embryos for analysis. Parallel studies in rodents for infertility treatments could be provided, including those on transgenerational disorders [127,131]. Additionally, research involving embryos generated from pluripotent stem cells (PSCs) replicates, in vitro, the processes that occur in embryonic development during a crucial period of observation and experimentation. This approach enables the modeling of the dynamics and stages of embryonic development.

Genetically edited iPSCs of large animals could enable researchers to understand the mechanisms of diseases, test potential therapeutic interventions, and assess the safety and efficacy contributing to pre-clinal trials [132,133,134]. The porcine species is an appealing model for investigating human genetic diseases because of the genetic similarity shared between the two species [132,135]. For the reproductive system, such as ovarian diseases, bovines are a suitable model [122,124]. Moreover, genetically edited iPSCs from both porcine and bovine species can contribute to veterinary medicine by improving reproductive traits and the development of resilient animals against diseases [133].

Hence, large animal iPSCs are valuable for understanding reproductive processes, modeling disorders, and developing potential therapeutic interventions. While species-specific differences must be considered, the translational impact of bovine iPSC research holds significant promise for advancing the field of reproductive medicine in humans.

## 5. Conclusions

Recent advances in mammalian embryology and stem cell biology have increased interest in the development of multicellular systems based on self-organization and tissue patterning. This emerging field aims to use PSCs to create organized multicellular models for numerous applications. Recent reproductive technology methods, such as the in vitro production of mature germ cells using IVG, have been used successfully in rodents; however, in other species, developing more efficient ways of generating these gametes and embryos is still necessary. The generation of embryo-like structures (or blastoids) has important and ambitious applications in translational medicine. Thus, the generation of these new technologies in large animals, such as bovines and pigs, is aimed at the improvement of animal production and agriculture through the generation of disease-resistant and genetically superior animals, genetically modified or not, in addition to accelerating the process in the generation of new individuals by reducing the interval between generations, as well as for the production of models for development studies and disease modeling.

## Figures and Tables

**Figure 1 jdb-12-00014-f001:**
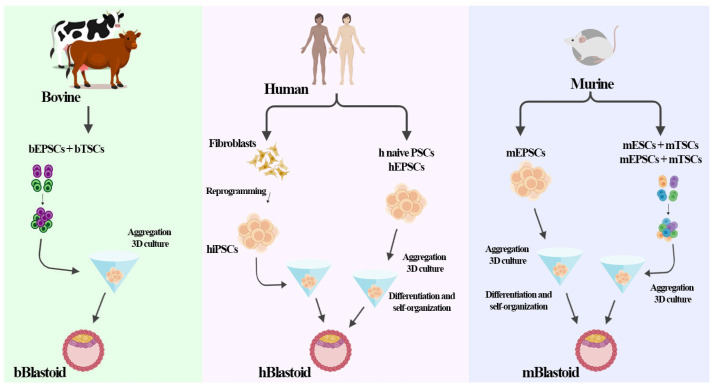
Formation of blastoids with different types of pluripotent stem cells. A schematic illustration shows the generation of blastoids in some mammalian species (cattle, mice, and humans). In bovines, blastoids were generated by aggregating different types of stem cells, EPSCs and TSCs. Human blastoids were generated by aggregating reprogrammed cells (iPSCs) or by the differentiation and self-organization of single-stem cells (naïve PSCs or EPSCs). Murine blastoids were generated by the differentiation and self-organization of single-stem cells (EPSCs) or by the aggregation of different types of stem cells (ESCs and TSCs or EPCs and TSCs).

**Figure 2 jdb-12-00014-f002:**
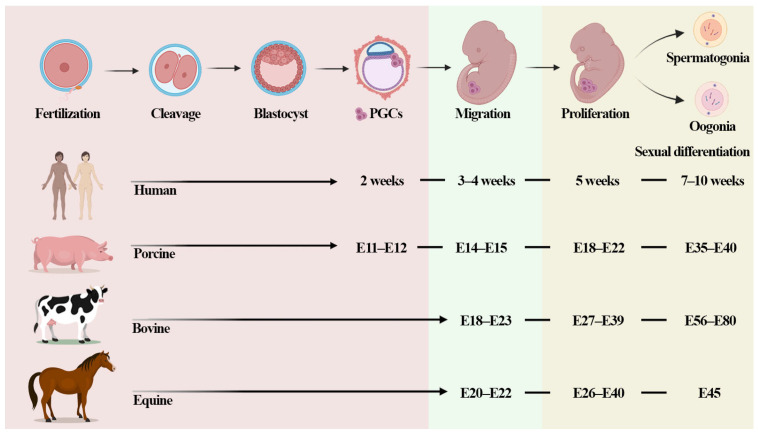
The development of primordial germ cells (PGCs) in different species. PGCs are specified in humans in the region of the amnion around 2 weeks; in porcine species, PGCs are specified in the posterior proximal epiblast around day 11–12; in bovines and horses, no studies have described when PGCs arise. The cells migrate to the genital ridge in humans at 3–4 weeks, in porcine species at 14–15 days, and in bovines at 18–23 days. PGCs arrive in the genital ridge and proliferation occurs in humans at 5 weeks, in porcine species at 18–22 days, in bovines at 27–39 days, and in equines at 26–40 days. Sexual differentiation was described in the embryos of humans at 7–10 weeks, in porcine species at 35–40 days, in bovines at 56–80 days, and in equines after 45 days of gestation.

**Figure 3 jdb-12-00014-f003:**
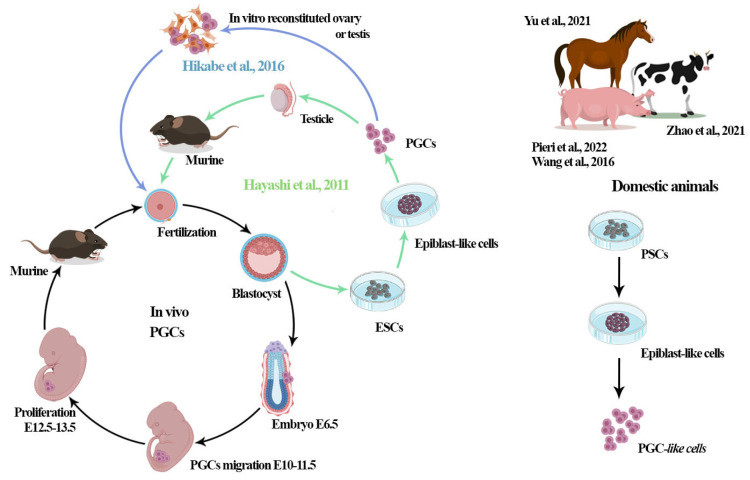
The development of new technologies and protocols, such as the production of PGCLCs and mature gametes through pluripotent stem cells (iPSCs and ESCs). The scheme shows the protocols described in mice by [12,40], displaying that PSCs can be induced into epiblast cell-like cells (EpiCLs) and then into PGC-like cells (PGCL-like or PGCLCs) with the capacity for complete spermatogenesis and oogenesis and the production of healthy offspring. In 2016, Hikabe et al. [41] described the generation of sperm-like and oogonia-like cells (OLCs) in vitro using the technology of the in vitro reconstitution of the ovary and testes in mice. Further studies have been based on the same protocol described by Hayashi et al. [12,98] with minor modifications in other animals, such as porcine, equine, and bovine species [38,39,45], reporting the generation of PGCLCs (PGC-like cells) from pluripotent stem cells (iPSCs or ESCs). Currently, there are no descriptions of embryos or offspring derived from these cells in these species.

**Figure 4 jdb-12-00014-f004:**
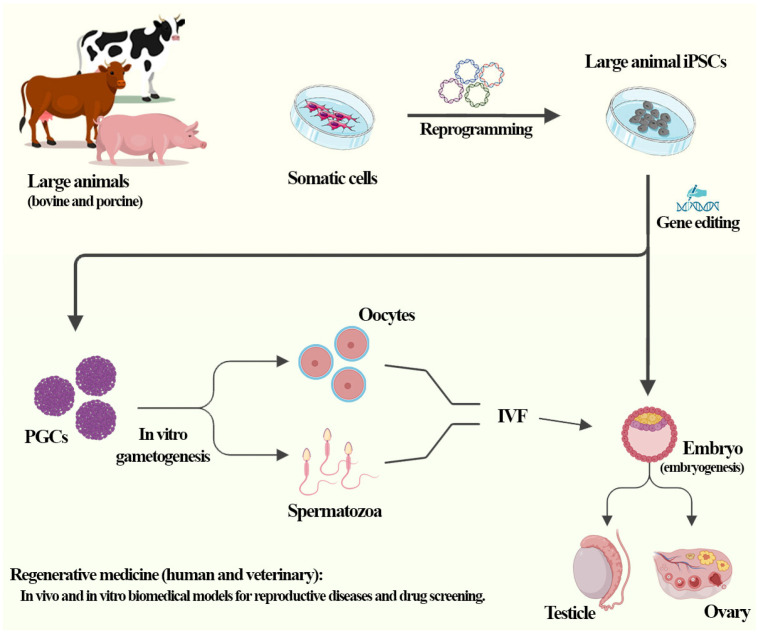
Large animal iPSCs (bovine and porcine) are used as a model to study embryogenesis, gametogenesis, reproductive diseases, and drug screening in human and veterinary regenerative medicine.

**Table 1 jdb-12-00014-t001:** History of in vitro embryo production in mammals.

Reference	Species	Description
[14]	Rabbit	The oocytes were matured in vivo, and the sperm was capacitated in the female reproductive tract. Fifteen live and healthy animals were born.
[15]	Mice	The oocytes were obtained from the female’s oviduct, and the sperm was collected from the uterus of a mated female. The fetuses were 17 days old.
[16]	Human	The oocyte was retrieved by laparoscopy, followed by in vitro fertilization. A healthy, normal child weighing 2.7 kg was born.
[17]	Porcine	The oocytes were punctured from the follicles and then transferred to the oviduct of the female, which was artificially inseminated. The embryos continued to develop normally for up to 25 days.
[18]	Bovine	The oocytes were surgically collected and fertilized in vitro, resulting in the birth of a 45 kg calf.
[19]	Equine	The oocytes were matured in vivo and recovered from the stimulated dominant follicle, and the sperm was treated with calcium ionophore A23187. The pregnancy was brought to term with the birth of two foals.

## Data Availability

Not applicable.

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
