# Peer review of "Emerging Contributions of Pluripotent Stem Cells to Reproductive Technologies in Veterinary Medicine"

_jdb, 2024, doi:10.3390/jdb12020014_

Round 1

Reviewer 1 Report

Comments and Suggestions for Authors

The Fratini de Castro et al manuscript entitled: “Emerging contributions of pluripotent stem cells to reproductive technologies in veterinary medicine” aims to provide an overview of recent developments in the veterinary stem cell field.

However, the work falls short in this respect for several reasons; large parts of Chapter 2 are outdated and refer to textbook knowledge of embryo production and transfer experiments dating back to the mid of last century; on several occasions the similarity of large animals physiology, embryology, etc is redundantly stressed, figs 3 and 4 are also redundant; and a number of cited approaches to generate gametes in vitro are euphemistically concluded as “and enabling, hopefully, the use this technique [sic] in other species.” /” …, the achievement of the mature gamete was not described…” / “…the generation of sperm-like cells…has yet not been described in this species.” / “…however these [sic] were not analyzed for feasibility or functionality.” (all of p9) , thus lacking a rigorous and critical review of the literature.

The English is hard to read, and the Reference list contains several erroneous citations and type setting errors.

The review is thus of very limited interest for a researcher in the field.

Comments on the Quality of English Language

poor

Reviewer 2 Report

Comments and Suggestions for Authors

In vitro gametogenesis/embryos from pluripotent stem cells (PSCs including ESCs, iPSCs, EPSCs, TSCs, etc.) is essential for basic medicine, stem cell biology, animal science, and veterinary medicine, especially to large mammals. The authors systematically discussed the history and the recent progress in this field by sorting relevent references. The four figures are very attractive and made the main text easily understood. Overall, the manuscript is well organized and written in fluent English, based on enough literature analysis. The following two issues need to be improved in the reviewer's eyes. 

1. The reference style in the legend of Figure 3 is different with the main text.

2. "Table 1 History of in vitro embryo production in mammals" is not very usfull, if the authors want to keep it, the "Descriptions" need to be shortened greatly.

Comments on the Quality of English Language

Quality of English needs to be improved.

Reviewer 3 Report

Comments and Suggestions for Authors

This is a detailed review on the production of in vitro embryos across multiple species using  variety of techniques. It is generally well written but would benefit from additional editing for English language. Some paragraphs are not currently clear as detailed below, and some areas would benefit from some additional summary tables to help the reader with the large amount of information that is provided in the text.

Line 41 – I think this should read “However, although the pre-implantation process….”

Line 82 – typo “suvh”

Line 91-92. Suggest to remove the second use of the word “first”

Table 1. Chang et al “joining the oocyte and sperm”. Please clarify what you mean by joining? Was the sperm injected? Co-cultured?

Table 1. Steptoe and Edwards. I think you mean 2700 g or 2.7 kg.

Line 108 – Please rephrase this sentence “Unlike IVF permissible species such as cattle, IVF in horses is difficult to establish and reproduce; currently, IVP is actually viable through ICSI. I cannot currently work out if you are saying you can make horse embryos through ICSI so then why is IVF hard to do in horses.

Line 149-151 – check tense used.

Figure 1. Please clarify the legend as it is a little complicated to follow (e.g. “or not”). In the bovine panel, two cell types are mentioned but 4 coloured cells are shown?

Section on new technologies – lines 136 onwards. Have any of these models been tested to see if live births can result? If not, why not? (obviously for ethical reasons in humans, but what about the other species?). Is there a time point at which human blastoids cannot be cultured beyond? This would give further support to the need to have models from other species.

Why would livestock species be better than mice or small animals for studying human development?

Section 2. A large amount of information on the old technologies in the different species is provided. Could this be summarised in a table?

Line 200 “In livestock, these mechanisms are very few elucidated yet.” Please rephrase for grammar.

Line 351-355. Please  revise this paragraph as it is not clear what exactly you are trying to state.

Line 395 20112?

Figure 3 legend – the dates given for the references in the legend do not all match the ones in the picture e..g Pieri et al 2021 or 2022? The legend refers to references that are not described in the picture (e.g. Hikabe 2016) which is confusing.

Line 358 – what species is this reference (108) – bovine?

Comments on the Quality of English Language

The manuscript is generally well written but would benefit from additional minor editing for English language in places.

Round 2

Reviewer 1 Report

Comments and Suggestions for Authors

The revision did not improve the paper, but the authors made minimal changes without addressing the concerns raised in the initial review, which is why the decision remains that the paper is not ready for publication in its current form.

Comments on the Quality of English Language

needs editing

Author Response

Please find the new answers in the attached file. We are sure we have properly addressed the questions and improved the manuscript. The manuscript was also corrected by a professional English service.
